# Trandolapril Attenuates Pro-Arrhythmic Downregulation of Cx43 and Cx40 in Atria of Volume Overloaded Hypertensive and Normotensive Rats

**DOI:** 10.3390/biom15101457

**Published:** 2025-10-15

**Authors:** Matúš Sýkora, Katarína Ondreják Andelová, Alexandra Mrvová, Barbara Szeiffová Bačová, Narcis Tribulová

**Affiliations:** Centre of Experimental Medicine, Institute for Heart Research, Slovak Academy of Sciences, 841 04 Bratislava, Slovakia; katarina.andelova@savba.sk (K.O.A.); mrvova.alexandra@gmail.com (A.M.); usrdbaca@savba.sk (B.S.B.)

**Keywords:** hypertensive TGR, normotensive HSD, ACF volume overload, left and right atria, connexin-40, connexin-43, trandolapril, losartan

## Abstract

Pressure overload in non-treated or resistant hypertension (HTN) increases the risk of heart failure (HF) as well as the occurrence of fatal ventricular arrhythmias and stroke-provoking atrial fibrillation (AF), while perturbed connexin-43 (Cx43) and Cx40 might be involved. In addition, kidney dysfunction may facilitate hemodynamic volume overload and congestive HF. We investigated the impact of volume overload on Cx43 and Cx40 in right and left heart atria of hypertensive pressure overloaded Ren-2 transgenic (TGR) strain and normotensive Hannover Sprague Dawley (HSD) rats, as well as the efficacy of renin–angiotensin blockade with trandolapril and losartan. Key novel findings revealed lower levels of Cx43 and Cx40 proteins in left as well as right heart atria in pressure overloaded hypertensive rats compared to normotensive rats. There was a significant decrease in Cx43 and Cx40 proteins due to volume overload in both atria of normotensive as well as hypertensive rats. Treatment with trandolapril increased Cx43 and Cx40 levels in right and left heart atria of normotensive as well as hypertensive volume overloaded rats. While losartan increased Cx43 and did not affect Cx40 in left and right heart atria of volume overloaded rats. Findings of this study point out that right heart atria of normotensive as well as hypertensive rats are more susceptible to volume overload comparing to the left heart atria. Trandolapril attenuated pro-arrhythmic downregulation of Cx43 and Cx40 in atria of volume overloaded normotensive as well as hypertensive rats. This fact as well as examining AF inducibility requires further investigation.

## 1. Introduction

Arterial hypertension (HTN)-induced hemodynamic pressure overload is a prevalent cardiovascular risk factor associated with the development of heart failure (HF), malignant cardiac arrhythmias, and adverse kidney events in both clinical and experimental settings [1,2,3,4]. Mechanical dysfunction and electrical instability aggravate over time and contribute significantly to morbidity and mortality [5], thereby highlighting the need for further cardioprotective investigation. Spontaneously hypertensive rats (SHR, a polygenic model imitating human primary HTN) and the hypertensive Ren-2 transgenic rats (TGR, a monogenic model of HTN) are experimental models imitating HTN associated with pressure overload [3,6]. Structural remodeling—mainly myocardial hypertrophy and interstitial collagen deposition—accompanied by disorders of electrical cardiomyocyte coupling at the connexin channels is considered a key factor in the deterioration of cardiac function and the increased vulnerability to arrhythmias [7,8]. On the other hand, renin–angiotensin–aldosterone system (RAAS) inhibition, resulting in the suppression of fibrosis, has been correlated with a reduction in arrhythmias [9]. Despite recent advances in cardiovascular pharmacotherapy, the prevention and treatment of resistant HTN and pulmonary arterial hypertension remain limited [10]. This highlights the need for novel, more effective, and safer pharmacological strategies.

Compared to pressure overload, much less is known about the impact of volume overload, which is typical in decompensated patients or those with renal disease/dysfunction [2,11]. In a rat model of acute decompensation due to volume overload induced by aorto-caval fistula (ACF), the condition—when combined with a hypertensive stimulus—progressively deteriorates over time into chronic congestive HF [12]. Myocardial stretch occurring during hemodynamic overload regulates caveolae-linked β_2_-adrenergic receptor (β_2_-AR) signaling [13], which contributes to the pathogenesis of HF. However, despite the high vulnerability to atrial fibrillation (AF), affecting approximately 60–80% of hypertensive patients [14], there remains a gap in comprehensive knowledge regarding atrial alterations. Chronic pressure or volume overload promotes AF [15,16,17] particularly through enhanced extracellular matrix accumulation and fibrosis, which contribute to the arrhythmogenic substrate. Hemodynamic overload of the atria is an important pathogenic factor in structural remodeling, including fibrosis [14,18]. The latter results from the stretch of overloaded cardiac myocytes, which directly activates fibroblasts [19]. Noteworthy, structural alterations accompanied by disturbances in the electrical coupling protein connexin-43 (Cx43) are associated with abnormal myocardial conduction, which is highly pro-arrhythmic [20], and contribute to the development of cardiac arrhythmias, including atrial fibrillation (AF). AF is further characterized by elevated levels of inflammatory biomarkers [21]. In contrast, improving intercellular electrical communication through Cx43 and Cx40 gap junction channels expressed in heart atria [22] alleviates AF [23].

However, it remains uncertain whether the use of specific antihypertensive drugs, such as those inhibiting the RAAS, can reduce pro-arrhythmic factors and thereby lower the risk of AF. Emerging antihypertensive drug classes offer new opportunities for managing HTN [2,24]. However, their long-term effects on cardiovascular, kidney, and metabolic outcomes as well as on arrhythmias, including AF, remain to be elucidated. This is particularly important given the rising prevalence of AF, which is becoming one of the major epidemics and public health challenges.

Based on the above clinical and preclinical findings, including our recently published data obtained from heart ventricles of volume overloaded rats [7], we aimed to investigate myocardial structure, enzyme histochemistry, and expression of the electrical coupling proteins Cx43 and Cx40 in the left and right atria of hypertensive as well as normotensive rats subjected to volume overload. In addition, we evaluated the efficacy of RAAS with trandolapril (an ACE inhibitor) and losartan (an angiotensin receptor blocker).

## 2. Materials and Methods

### 2.1. Experimental Design

This study was performed using male heterozygous hypertensive Ren-2 transgenic rats [(mREN2) 27, TGR; n = 40] and normotensive Hannover Sprague Dawley rats (HSD; n = 40), housed under standard laboratory conditions (22 ± 1 °C, 12:12 h light/dark cycle), with ad libitum access to chow and water. Congestive heart failure (HF) due to volume overload was induced in 8-week-old animals by creating an aortocaval fistula (ACF) using an 18-gauge needle (1.2 mm in diameter), followed by sealing with cyanoacrylate adhesive [7,11]. ACF induction was confirmed by the presence of pulsatile blood flow in the inferior vena cava. Sham-operated rats served as controls. Five weeks after ACF-induced high-volume HF in normotensive and hypertensive rats (as documented in previous study [11]) examining rat heart ventricles and kidney a pharmacological intervention was initiated for a duration of 15-week. Rats received daily either the angiotenzin II type 1 (AT_1_) receptor blocker, losartan (200 mg/L), or the angiotenzin-converting enzyme inhibitor (ACEi) trandolapril (6 mg/L) in drinking water. Age-matched untreated rats were used for comparison. At 28 weeks of age, animals were euthanized by decapitation. Blood serum, as well as the left and right ventricles and atria, were collected. Biometric parameters were recorded, and heart tissue samples were snap-frozen in liquid nitrogen and stored at −80 °C.

All procedures were approved by the Animal Care and Use Committee of the Institute for Clinical and Experimental Medicine (project number 50/2017) and adhered to Directive 2010/63/EU and ARRIVE 2.0 guidelines as previously reported [7,11].

### 2.2. Serum Biomarker Assessment

Atrial natriuretic peptide (ANP), a biomarker of heart failure, was quantified in serum using a commercial ELISA kit (RAB00385, Sigma-Aldrich, St. Louis, MO, USA) according to the manufacturer’s instructions. Serum samples and standards were incubated in antibody-coated wells, followed by incubation with HRP-conjugated streptavidin and TMB substrate. Absorbance was measured at 450 nm using a microplate reader (Synergy H1, BioTek, Winooski, VT, USA), and ANP concentrations (pg/mL) were calculated based on the standard curve.

Lipid peroxidation was assessed by measuring thiobarbituric acid reactive substances (TBARS) using a modified spectrophotometric method [25]. Serum samples or standards were treated with trichloroacetic acid and TBARS reagent, incubated at 100 °C for 70 min, and subsequently extracted with a mixture of n-butanol and pyridine (14:1). After centrifugation (5000× *g*, 10 min), the absorbance of the organic phase was measured at 535 nm using a microplate reader (Synergy H1, BioTek, Winooski, VT, USA). Malondialdehyde (MDA) concentrations were determined from a standard curve prepared with tetrabutylammonium MDA salt.

Zymographic analysis of MMP-2 gelatinolytic activity was performed as previously described by [7]. In brief, protein samples were subjected to SDS-PAGE in gelatin-containing gels under non-reducing conditions. After electrophoresis, gels were incubated in substrate buffer and subsequently stained with Coomassie Brilliant Blue. Gelatinolytic activity was visualized as clear bands against a dark background.

TIMP-2 levels were measured using a Rat TIMP-2 ELISA kit (RAB1156, Sigma-Aldrich, St. Louis, MO, USA) according to the manufacturer’s instructions. Samples were sequentially incubated with a biotinylated detection antibody, HRP-conjugated streptavidin, and TMB substrate. Absorbance was measured at 450 nm using a microplate reader (Synergy H1, BioTek, Winooski, VT, USA), and TIMP-2 concentrations (pg/mL) were calculated from a standard curve.

### 2.3. Microscopic Examination of Left and Right Heart Atrial Tissue

Conventional hematoxylin-eosin and Masson’s trichrome staining were performed on 10 μm-thick cryosections of the right and left atria to evaluate structural alterations.

Catalytic enzyme histochemistry was performed on 10 μm-thick cryosections of the right and left atria to assess the arterial and venous capillary network in situ by determining the activity of endothelial alkaline phosphatase (AP, EC 3.1.3.1), a marker of arterial capillaries, and dipeptidyl peptidase-4 (DPP4, EC 3.4.15.4), a marker of endothelial venous capillaries [26,27]. For the detection of AP activity, tissue sections were incubated in a buffered medium containing sodium β-glycerophosphate as the substrate, resulting in blue staining of arterial capillaries. DPP4 activity was visualized using glycyl-L-proline-4-methoxy-β-naphthylamide as the specific substrate, resulting in red staining of venous capillaries.

The activity of glycogen phosphorylase (GP, EC 2.4.1.1), the enzyme involved in myocardial glycogen breakdown, was determined according to [26]. Cryosections (10 μm thick) of the right and left atria were incubated in a buffer solution containing glycogen as a primer and glucose-1-phosphate as the glycosyl group donor. The product of the reaction was a purple-colored final precipitation.

Histochemical activities of the mitochondrial enzymes succinate dehydrogenase (SDH, EC 1.3.5.1) and β-hydroxybutyrate dehydrogenase (β-HBDH, EC 1.1.1.30), involved in myocardial energy metabolism, were examined according to the method of [26]. Cryosections (10 μm thick) of the right and left atria were incubated in a phosphate-buffered medium containing succinate and β-hydroxybutyrate as substrates, NAD as a coenzyme, and nitroblue tetrazolium chloride as a hydrogen acceptor, resulting in the formation of blue formazan granules indicating enzyme activity. All images were acquired and examined using a Zeiss Apotome 2 light microscope (Carl Zeiss Microscopy GmbH, Jena, Germany).

To assess the expression and localization of connexin-43 (Cx43) in the atrial myocardium, immunofluorescence staining was performed on 10 μm-thick cryosections of the right and left atria, following the protocol described by [28]. Tissue sections were incubated with a primary mouse anti-Cx43 antibody (1:700; MAB3068, CHEMICON International, Inc., Temecula, CA, USA), followed by a FITC-conjugated secondary antibody specific to mouse IgG (1:1000; Jackson ImmunoResearch Laboratories, West Grove, PA, USA). Immunofluorescence was examined using a Zeiss Apotome 2 flourescent microscope (Carl Zeiss Microscopy GmbH, Jena, Germany).

Quantitative image analysis of microscopic images was not assessed because the number of hypertensive rats suffering from volume overload HF was reduced due to mortality.

### 2.4. Western Blot Analysis of Left and Right Atrial Tissue

About 100 mg of frozen left and right heart atrium was homogenized in ice-cold lysis buffer containing 180 mmol/L KCl, 4 mmol/L EDTA, pH 7.4. A protease inhibitor cocktail (Sigma-Aldrich, St. Louis, MO, USA; #P8340) was also added to prevent protein degradation. Protein extracts were mixed with Laemmli sample buffer and heated prior to electrophoresis. Equal amounts of protein (6–60 µg per lane) were separated on 10% SDS–polyacrylamide gels (Mini-Protean TetraCell, Bio-Rad, Hercules, CA, USA) at a constant voltage of 90 V and transferred onto 0.2 µm nitrocellulose membranes (Advantec, Tokyo, Japan). Membranes were blocked in 5% low-fat milk (*w*/*v* in TBS-T) for 4 h at room temperature and incubated overnight at 4 °C with the appropriate primary antibodies: anti-total Cx43—1:5000, C6219, Sigma-Aldrich, MI, USA; anti-Cx40—1:2000, sc-365107, Santa Cruz Biotechnology, Dallas, TX, USA; anti-PKCε—1:1000, sc-214, Santa Cruz Biotechnology, TX, USA; anti-PKCδ—1:1000, sc-213, Santa Cruz Biotechnology, Dallas, TX, USA; anti-MMP-2—1:1000, sc-10736, Santa Cruz Biotechnology, Dallas, TX, USA; anti-Galectin-3—1:1000, #89572, Cell Signaling Technology, Denver, CO, USA; anti-ADAMTS5—1:500, ab41037, Abcam, Cambrige, UK; anti-GAPDH—1:1000, sc-25778, Santa Cruz Biotechnology, TX, USA. After washing in TBS-T, membranes were incubated with horseradish peroxidase-conjugated secondary antibodies: anti-rabbit antibody: 1:2000, 7074S, Cell Signaling Technology, Denver, CO, USA or anti-mouse antibody: 1:2000, 7076C, Cell Signaling Technology, Denver, CO, USA, for 1–1.5 h at room temperature. Protein bands were visualized using enhanced chemiluminescence (ECL) and quantified by densitometric analysis using Carestream Molecular Imaging Software (version 5.0; Carestream Health, New Haven, CT, USA). GAPDH was used as a loading control for protein normalization. All steps were previously described [2,29].

### 2.5. Statistical Evaluation

The Kolmogorov–Smirnov test was used to verify the normality of data distribution. Group differences were analyzed using one-way analysis of variance (ANOVA) followed by Bonferroni’s multiple comparison test. Results are presented as means ± standard deviations (SD), and a *p*-value of less than 0.05 was considered statistically significant. All statistical analyses were performed using GraphPad Prism 8.0.1.

## 3. Results

### 3.1. Biometric and Biochemical Parameters

As shown in Table 1, body weight (BW) and heart weight (HW) were significantly higher in hypertensive TGR strain compared to normotensive HSD rats. Volume overload induced by ACF significantly increased the HW, as well as the weight of left (LVW) and right ventricles (RVW), in both HSD and TGR, while BW was reduced in TGR. For technical reasons, it was not possible to register the right and left atrial weights in this experiment. Eight weeks of ACEi treatment attenuated the effects of volume overload and reduced LVW and RVW in both HSD and TGR strain, with a concomitant increase in BW in TGR. These biometric parameters were not reduced in volume-overloaded HSD rats treated with ARB, whereas ARB treatment did attenuate them in volume-overloaded TGR. Due to technical reasons atrial weights were not possible to register in this experiment. Serum ANP and TBARS levels were significantly increased due to volume overload in both HSD and TGR rats; however, ANP did not change in HSD rats with ACEi treatment, and TBARS levels were not altered after ARB treatment. No significant alterations in serum MMP-2 were observed among the rat groups, except for a significant reduction in TGR treated with ACEi. Serum TIMP-2 levels were significantly increased in atria of volume-overloaded HSD rats and in TGR rats treated with ACEi. These data were previously published in a papers focused on heart ventricles [7] and renal functions [11], whereby HF development was monitored and confirmed by Doppler sonography.

### 3.2. Microscopic Findings of Left and Right Atrial Tissue of Volume-Overloaded Normotensive and Hypertensive Rats

Microscopic alterations demonstrated on images point out the differences between right and left heart atria including responsiveness to volume overload induced conges-tive heart failure in normotensive HSD rats and hypertensive TGR, as well as impact of treatment with trandolapril and losartan. See Figure 1, Figure 2, Figure 3 and Figure 4.

Volume overload resulted in increased number of polymorphonuclears in right atria of normotensive rats and to lesser extent in TGR, while treatment with trandolapril suppressed their incidence in both normotensive and hypertensive rat strain (Figure 1).

As shown in Figure 2, an increase in interstitial collagen deposition due to volume overload was evident in the left and namely in the right atria of normotensive rats, which was attenuated by treatment with angiotensin-converting enzyme inhibitor trandorapril or angiotensin receptor blockers losartan. Peri-arterial fibrosis was observed in the let atria of hypertensive rats that was not further exacerbated by volume overload. Whilst the right atria exhibited massive interstitial and perivascular collagen accumulation. Treatment with trandorapril partially suppressed extracellular matrix formation.

Volume overload increased the density of arterial capillaries, particularly in the right atria of normotensive rats, along with elevated enzyme activity in the epicardium as demonstrated in Figure 3. These alterations persisted despite the treatment. Of interest, the density of the arterial capillary network exhibiting alkaline phosphatase activity was higher in the right versus left atria of hypertensive rats. Volume overload reduced the capillary network in the right atria of hypertensive rats while increasing it in normotensive rats. Treatment with trandolapril partially attenuated volume overload-induced alterations.

Dipeptidyl peptidase-4 activity in the venous capillary network of the left and right atria of experimental rats demonstrated in Figure 4, revealed a higher density in the left versus right atria in normotensive as well as hypertensive rats. Neither volume overload nor treatment affected this pattern.

Immunolabeling of connexin-43 revealed its dominant localization at the intercalated disks of the cardiomyocytes in the left and right atria of normotensive rats, as demonstrated in Figure 5. In contrast, lower myocardial density of Cx43 and its mis-localization to the lateral cardiomyocyte membranes were detected in atria of hyper-tensive rats. Volume overload reduced Cx43 in right and left atria of normotensive rats and to lesser extent in hypertensive rats, while abnormal lateral localization persisted. Treatment with trandolapril or losartan increased Cx43 and attenuated its mis-localization in atria of volume overloaded normotensive and partially in hypertensive rat heart.

### 3.3. Findings on Examined Protein Levels in the Left and Right Atria of Volume-Overloaded Normotensive and Hypertensive Rats

As shown in Figure 6, hypertensive TGR exhibited lower levels of Cx43 (Figure 6A) and Cx40 (Figure 6B) in both the left and right atria compared to normotensive HSD rats. A pronounced decrease in Cx43 and Cx40 protein levels was observed in both atria of normotensive HSD and hypertensive TGR strain due to volume overload. Treatment with an ACEi increased Cx43 and Cx40 levels in both atria of volume overload normotensive HSD and hypertensive TGR. In contrast, treatment with an ARB increased Cx43 but did not affect Cx40 levels in either atrium of volume-overloaded rats.

PKCε is involved in the phosphorylation of Cx43 and Cx40, which activates their function. As shown in Figure 7A, compared to normotensive HSD rats, PKCε levels were lower in both the left and right atria of hypertensive TGR. Volume overload significantly reduced PKCε protein levels only in the left atrium of normotensive HSD rats. Treatment with an ACEi increased PKCε levels in both atria of volume-overloaded normotensive HSD and hypertensive TGR strain.

PKCδ is involved in pro-hypertrophic and pro-apoptotic signaling. As shown in Figure 7B. PKCδ protein levels were significantly increased due to volume overload in both atria of normotensive HSD rats and hypertensive TGR. Treatment with an ACEi reduced PKCδ levels in the left and right atria of volume-overloaded hypertensive TGR only and did not alter the levels in the atria of normotensive rats.

Proteins involved in extracellular matrix remodeling are shown in Figure 8.

MMP-2 levels (Figure 8A) were significantly reduced in both the left and right atria due to volume overload, but only in normotensive HSD rats. Treatment with an ACEi increased MMP-2 protein levels in the left atrium of volume-overloaded TGR rats.

Galectin-3 levels (Figure 8B) were increased in the left and to a lesser extent in the right atria of normotensive and hypertensive rats in response to volume overload, regardless of treatment. Interestingly, Galectin-3 was significantly reduced in the right atria following treatment of volume-overloaded normotensive rats with either an ACEi or an ARB.

ADAMTS levels (Figure 8C) were not affected by volume overload in the left atrium, but were decreased in the right atrium of normotensive HSD rats. Notably, treatment with either an ACEi or an ARB increased ADAMTS proteins in the left atrium of volume-overloaded TGR rats.

## 4. Discussion

Compared to the heart ventricles, much less is known about atria under various pathological conditions, despite the fact that atrial fibrillation (AF), the most common arrhythmia and a major risk factor for ischemic stroke, is permanently increasing [30]. Moreover, AF and heart failure (HF) are two commonly coexisting conditions that are frequently encountered in clinical practice [14,31]. In this context, our aim was to investigate myocardial structure, enzyme histochemistry, and the expression of electrical coupling proteins Cx43 and Cx40 in the left and right atria of male hypertensive Ren-2 transgenic rats (TGR) and normotensive Hannover Sprague Dawley (HSD) rats subjected to volume overload to induce congestive HF. In addition, the efficacy of RAAS blockade using trandolapril (an ACE inhibitor) and losartan (an angiotensin receptor blocker) was evaluated.

According to the recorded biometric parameters (Table 1), which were previously published in papers focused on heart ventricles [7] and renal functions [11], hypertensive rats exhibited higher body weight (BW) and heart weight (HW) compared to normotensive rats. Volume overload increased HW, left ventricular weight (LVW), and right ventricular weight (RVW) in both rat strains, while it reduced BW in hypertensive rats. When exposed to identical volume overload, both ventricles display similar upregulation of stress and metabolic markers [32]. Relatively larger response of volume overloaded in the right ventricle compared to the left may be caused by concomitant pulmonary hypertension [33]. However, there is still minimal data on the adaptation of the right ventricle to pressure and volume overload and the transition to right ventricular failure. No evidence supports right ventricular chamber-specific regulation of protein expression in response to volume overload. Although on the cellular level, the remodeling responses of the right and left ventricles to pressure overload are largely similar, there are several key differences: the stressed right ventricle is more susceptible to oxidative stress, has a reduced angiogenic response, and is more likely to activate cell death pathways than the stressed left ventricle [33]. The need to better understand the molecular mechanisms allows for the development of right ventricle-specific heart failure therapeutics. This would be much appreciated in managing pulmonary hypertension. It is supposed that atrial dilatation due to volume overload may increase their weight as well. However, biometric parameters of atria are missing in the current study because their registration during harvesting of atrial tissue was not possible. Nonetheless, treatment of volume overload with trandolapril improved these attenuated altered ventricular biometric ventricular parameters in hypertensive rat strain [7]. While treatment with losartan did not affect biometric parameters in volume-overloaded normotensive rats, unlike in hypertensive rats. It might be explained by different mechanisms of their antihypertensive effects. Mean arterial pressure was higher in hypertensive versus normotensive rats, and volume overload significantly reduced this parameter only in hypertensive rats. Treatment with either trandolapril or losartan suppressed this parameter only in hypertensive volume-overloaded rats, as previously published [11]. In addition, cardiac output was significantly increased due to volume overload in normotensive as well as hypertensive rats, and even more after treatment with losartan. In addition, treatment reduced serum levels of atrial natriuretic peptide (ANP), thiobarbituric acid reactive substances (TBARS), and tissue inhibitor of metalloproteinases-2 (TIMP-2), all of which were elevated due to volume overload in the atria of both rat strains. Matrix metalloproteinases (MMPs), a family of proteolytic enzymes, and TIMP regulate the extracellular matrix turnover in cardiac tissue and are involved in cardiac fibrosis [34]. Inhibitors of matrix MMP 2 may minimize cardiac structural and functional alterations, thereby fighting heart failure [35]. We detected that volume overload decreased MMP 2 protein level in both atria of normotensive as well as hypertensive rat strain (Figure 8), while only treatment with trandolapril abolished this decrease in left atria of hypertensive strain. An increase in collagen deposition due to volume overload was detected in the right atria of normotensive rats, which was attenuated by treatment with either trandolapril or losartan, as shown in Figure 2. In the left atria of hypertensive rats, periarterial and interstitial fibrosis was not further aggravated by volume overload, in contrast to the right atrium, where interstitial collagen deposition was increased. Treatment partially attenuated extracellular matrix alterations in atria of volume-overloaded hypertensive rats, likewise in the ventricles, as we previously reported in the left ventricle [7]. Moreover, as shown in microscopic images in Figure 1, treatment with trandolapril or losartan reduced the abundance of polymorphonuclear cells induced by volume overload, particularly in the right atrium of normotensive rats. Compared to normotensive rats, the hypertensive strain exhibited more polymorphonuclear cells in the left atrium, with a further increase observed in the right atrium following volume overload. This increase was attenuated by trandolapril. Polymorphonuclears, in particular myeloid and lymphoid-derived immune cells such as macrophages, neutrophils, T-cells, and B-cells reflecting inflammation, are very active and may be involved in arrhythmia pathophysiology [36]. This issue requires further study regarding malignant cardiac arrhythmia prevention.

Adaptive changes in the myocardial capillary network were examined by in situ activity of endothelial alkaline phosphatase (Figure 3), a marker of arteriolar capillaries, and endothelial dipeptidyl peptidase-4 activity (Figure 4), a marker of the venular capillary network [27,37]. Volume overload induced a higher density of arteriolar capillaries, particularly in the right atrium of normotensive rats, along with increased enzyme activity in the epicardium. These qualitative changes were not affected by treatment with either trandolapril or losartan. In hypertensive rats, arteriolar capillaries were more abundant in the right compared to the left atrium, and their density was enhanced by volume overload in the left while reduced in the right atrium. There was no apparent effect of treatment. Venular capillary activity, indicated by dipeptidyl peptidase-4 red colored reaction, was higher in the left atrium compared to the right in both normotensive and hypertensive rats, regardless of intervention. Neither volume overload nor treatment with trandolapril or losartan altered this pattern. Of interest, the involvement of dipeptidyl peptidase-4 in the regulation of mitochondrial function and oxidative stress in cardiomyocytes has been recently reported [38] as well as in the development of cardiac fibrosis and atrial arrhythmias [39]. It should be emphasized that coronary microvascular dysfunction across the spectrum of heart pathologies contributes to the development and progression of HF [40].

Atrial energetic metabolism was assessed by in situ activity of cytoplasmic glycogen phosphorylase (Appendix A), mitochondrial succinic dehydrogenase (Appendix A), and β-hydroxybutyrate dehydrogenase (Appendix A). As shown in the supplementary data, no alterations indicating atrial ischemia were observed regardless of rat strain or volume overload conditions.

The most interesting novel findings of this study include alterations in the expression of the electrical coupling proteins Cx43 (Figure 6A) and Cx40 (Figure 6B). Compared to the normotensive rat strain, hypertensive rats exhibited lower levels of Cx43 and Cx40 in the left and right atria. Volume overload significantly suppressed the expression of Cx43 and Cx40 in both atria, regardless of rat strain. Changes in Cx43 expression were accompanied by altered localization, specifically a redistribution from intercalated disks to lateral cardiomyocyte membranes, as demonstrated in Figure 5, which impairs electrical conduction and is considered proarrhythmic [3,22,41]. It appears that atrial remodeling due to volume overload, characterized by structural changes and downregulation of Cx43 and Cx40, may impair myocardial electrical conduction and contribute to an increased propensity for AF [22], likewise in humans with congestive HF [42]. On the other hand, treatments, particularly with trandolapril, attenuated the downregulation of Cx43 and Cx40 in both the right and left atria, likewise previously found in the heart ventricles of volume-overloaded rats [7]. Trandolapril also reduced proarrhythmic electrical remodeling and mortality, indicating its beneficial therapeutic effects in volume overload HF [8]. This suggests pleiotropic antiarrhythmic effects, whose mechanisms require further investigation. Of interest, the multi-ion channel inhibitor acehytisine has shown benefits in preventing and terminating persistent AF in dilated atria caused by volume overload [17].

In addition, trandolapril increased PKCε protein levels in both right and left heart atria (Figure 7A), which were suppressed by volume overload in both normotensive and hypertensive rat strains. Considering that phosphorylation of Cx43 and Cx40 by PKCε is involved in the modulation of electrical conduction, it may therefore be expected to influence arrhythmogenesis [43]. However, treatment did not affect the volume overload-induced elevation of pro-hypertrophic PKCδ in either the left or right heart atria (Figure 7B), regardless of rat strain.

Pro-arrhythmic ventricular myocardial fibrosis during the progression of HF has been associated with increased MMP-2 levels [35]. In contrast, HF due to volume overload reduced MMP-2 protein levels in both left and right atria of normotensive rats (Figure 8A), while trandolapril unexpectedly increased MMP-2 in the left atria of hypertensive rats. Galectin-3 has emerged as a critical mediator involved in cardiac inflammation, vascular remodeling, and fibrotic processes [44]. Galectin-3 protein levels were increased in the left heart atria due to volume overload, regardless of treatment. Interestingly, Galectin-3 levels (Figure 8B) were significantly reduced in the right heart atria of volume-overloaded normotensive rats treated with either trandolapril or losartan. In contrast to the left atria, Galectin-3 was significantly increased in the right heart atria of the TGR strain. ADAMTS5 is a large extracellular matrix proteoglycan involved in extracellular matrix remodeling [45]. ADAMTS5 protein levels (Figure 8C) were not affected by volume overload in the left heart atria but were altered in the right atria of normotensive HSD rats. Of interest, treatment with either trandolapril or losartan increased ADAMTS protein levels in the left atria of volume-overloaded TGR strain.

Finally, it should be noted that the difference in action of Trandolapril and Losartan is primarily associated with their mechanisms of action. Trandolapril, an ACE inhibitor, inhibits the activity of the angiotensin-converting enzyme, a key component of the renin-angiotensin system that converts angiotensin I into angiotensin II. Thus, ACE inhibitors lead to a reduction in circulating levels of angiotensin II, which can itself directly activate or inactivate pathways of interest. For example, angiotensin II is involved in the down-regulation of Cx43 [46,47]. In contrast, Losartan, an angiotensin receptor blocker (ARB), does not reduce the level of circulating angiotensin II; it acts only by blocking the AT_1_ receptors. Any non-receptor-mediated functions of angiotensin II remain unchanged.

## 5. Conclusions

Novel findings of this study indicate that the right atria of normotensive HSD as well as hypertensive TGR are more susceptible to volume overload compared to the left atria. The downregulation of Cx43 and Cx40 caused by volume overload strongly suggests their participation in the development of atrial fibrillation, as these gap junction proteins are essential for electrical coupling and action potential propagation among cardiomyocytes. Noteworthy, trandolapril attenuated the pro-arrhythmic downregulation of Cx43 and Cx40 in the atria of volume-overloaded normotensive and hypertensive rats, while losartan was less efficient. Based on these findings, further studies should investigate the impact of volume overload on the incidence of atrial fibrillation in both normotensive and hypertensive rats, as well as explore the molecular mechanisms underlying Cx43 and Cx40 upregulation by trandolapril.

### Limitations of This Study

The main limitation of this study is the lack of examination of electrocardiographic parameters and testing propensity to atrial arrhythmias, as well as investigation of proposed antiarrhythmic mechanisms of pharmacotherapy that require further research and investigations. Another limitation of this study is the absence of quantitative analysis for microscopic findings. As the conclusions are based on representative images, they should be interpreted with caution. Further studies, including quantitative assessment, are required to validate these observations.

## Figures and Tables

**Figure 1 biomolecules-15-01457-f001:**
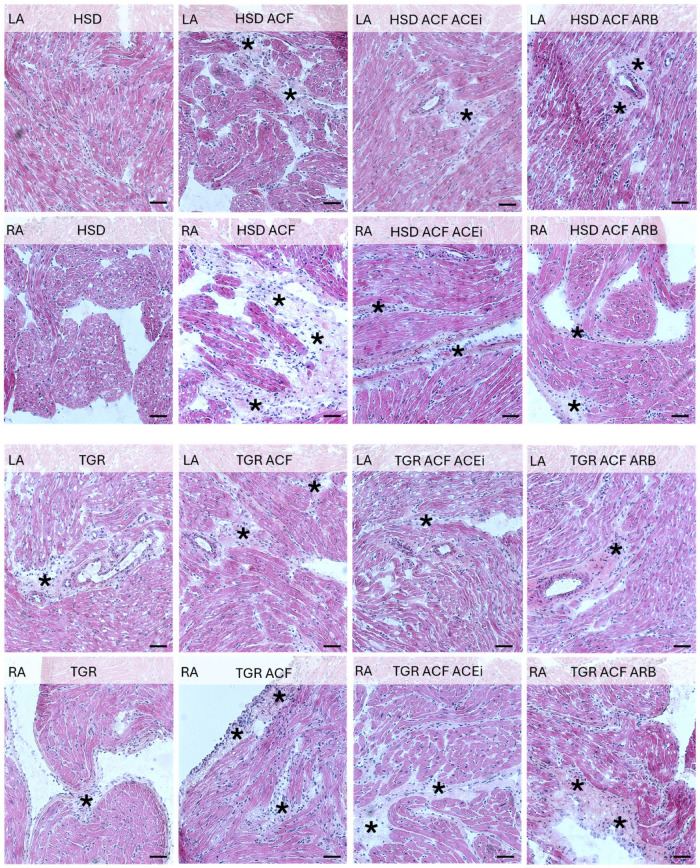
Haematoxylin-eosin staining of the left (LA) and right atria (RA) of experimental rats. Abundant polymorphonuclear cells (asterisk) are present in the RA of normotensive HSD rats due to ACF-induced volume overload, which was attenuated by treatment with ACE inhibitors (ACEi) and angiotensin receptor blockers (ARB). Compared to HSD rats, the atria of hypertensive TGR exhibit a higher number of polymorphonuclear cells, with volume overload further enhancing their presence, particularly in the RA. This effect was attenuated by ACEi treatment. HSD—Hannover Sprague Dawley rats; TGR—Ren-2 transgenic rats; ACF—aortocaval fistula, surgical model of volume overload; ACEi—treatment with the angiotensin-converting enzyme inhibitor, trandolapril; ARB—treatment with an angiotensin II type 1 (AT_1_) receptor blocker, losartan. Scale bar: 100 μm.

**Figure 2 biomolecules-15-01457-f002:**
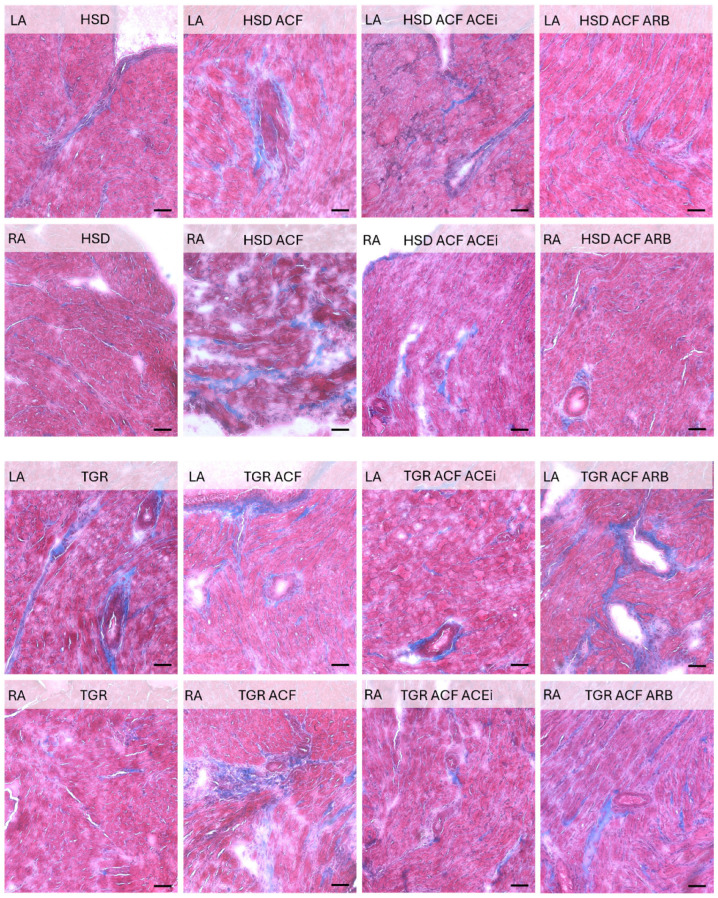
Masson trichrom staining of interstitial and periarterial collagen fibers (blue color) in left (LA) and right atria (RA) of experimental rats. Cardiomyocytes are stained pink. An increase in collagen deposition due to ACF-induced volume overload is evident in the left (LA) and especially in the right atria (RA) of normotensive HSD rats, which was attenuated by treatment with ACE inhibitors (ACEi) or angiotensin receptor blockers (ARB). In hypertensive TGR rats, the LA showed baseline fibrosis that was not further exacerbated by volume overload, in contrast to the RA, which exhibited massive interstitial collagen accumulation. Treatment partially suppressed extracellular matrix alterations. HSD—Hannover Sprague Dawley rats; TGR—Ren-2 transgenic rats; ACF—aortocaval fistula, surgical model of volume overload; ACEi—treatment with the angiotensin-converting enzyme inhibitor, trandolapril; ARB—treatment with an angiotensin II type 1 (AT_1_) receptor blocker, losartan. Scale bar: 100 μm.

**Figure 3 biomolecules-15-01457-f003:**
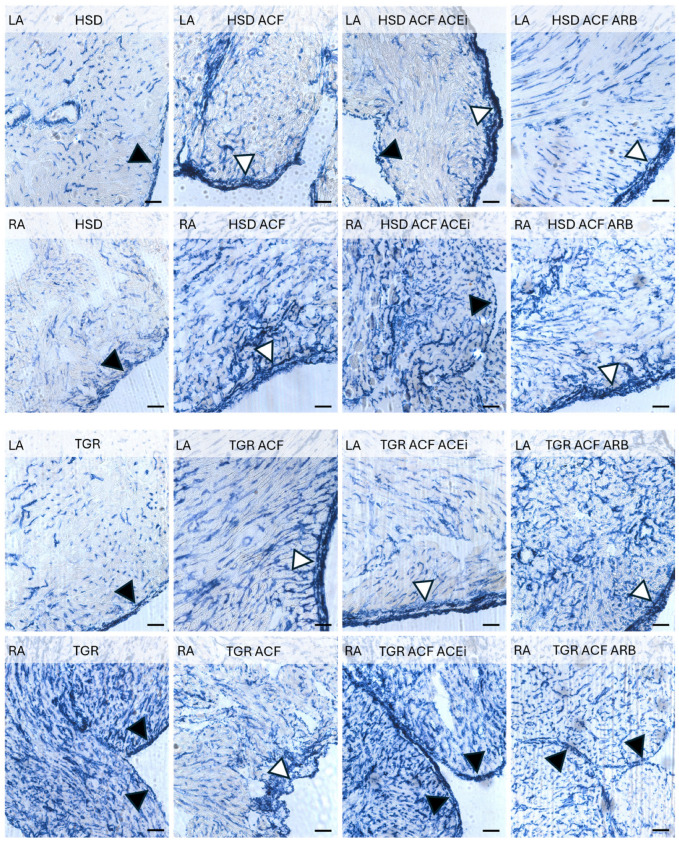
Alkaline phosphatase activity in the arterial capillary network (blue color) in the right (RA) and left atria (LA) of experimental rats. ACF-induced volume overload increased the density of arterial capillaries, particularly in the RA of normotensive HSD rats, along with elevated enzyme activity in the epicardium (white triangle), while the endocardium is highlighted with black triangles. These qualitative changes persisted despite treatment. In hypertensive TGR rats, the density of alkaline phosphatase-positive capillaries was higher in the RA compared to the LA and further enhanced in the LA following ACF. Treatment had no apparent effect. HSD—Hannover Sprague Dawley rats; TGR—Ren-2 transgenic rats; ACF—aortocaval fistula, surgical model of volume overload; ACEi—treatment with the angiotensin-converting enzyme inhibitor, trandolapril; ARB—treatment with an angiotensin II type 1 (AT_1_) receptor blocker, losartan. Scale bar: 100 μm.

**Figure 4 biomolecules-15-01457-f004:**
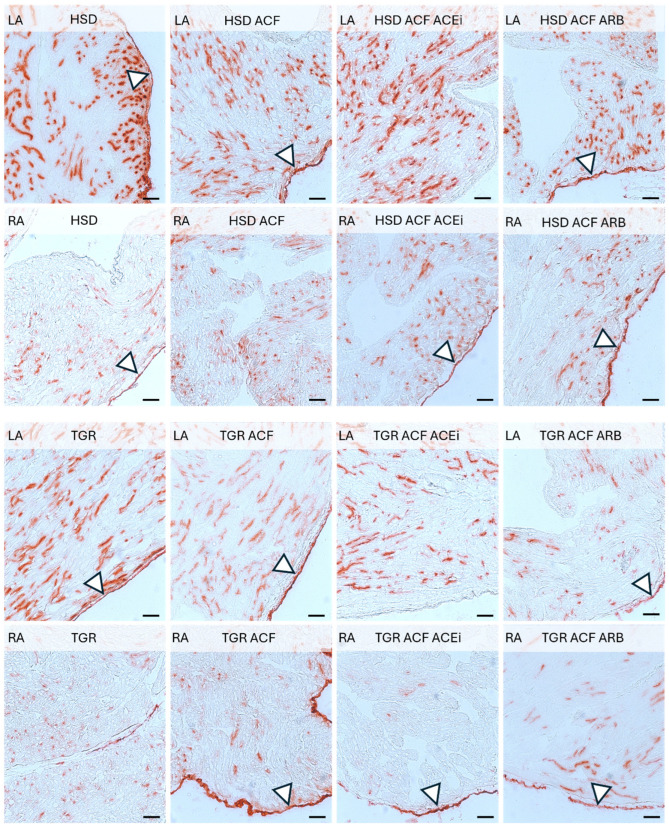
Dipeptidyl peptidase-4 (DPP4) activity (red color) in the venous capillary network of the left (LA) and right atria (RA) of experimental rats. Note the lower density and activity of DPP4 in the RA compared to the LA in both normotensive and hypertensive rats, regardless of intervention. White triangles indicate DPP4 activity in the epicardial region. HSD—Hannover Sprague Dawley rats; TGR—Ren-2 transgenic rats; ACF—aortocaval fistula, surgical model of volume overload; ACEi—treatment with the angiotensin-converting enzyme inhibitor, trandolapril; ARB—treatment with an angiotensin II type 1 (AT_1_) receptor blocker, losartan. Scale bar: 100 μm.

**Figure 5 biomolecules-15-01457-f005:**
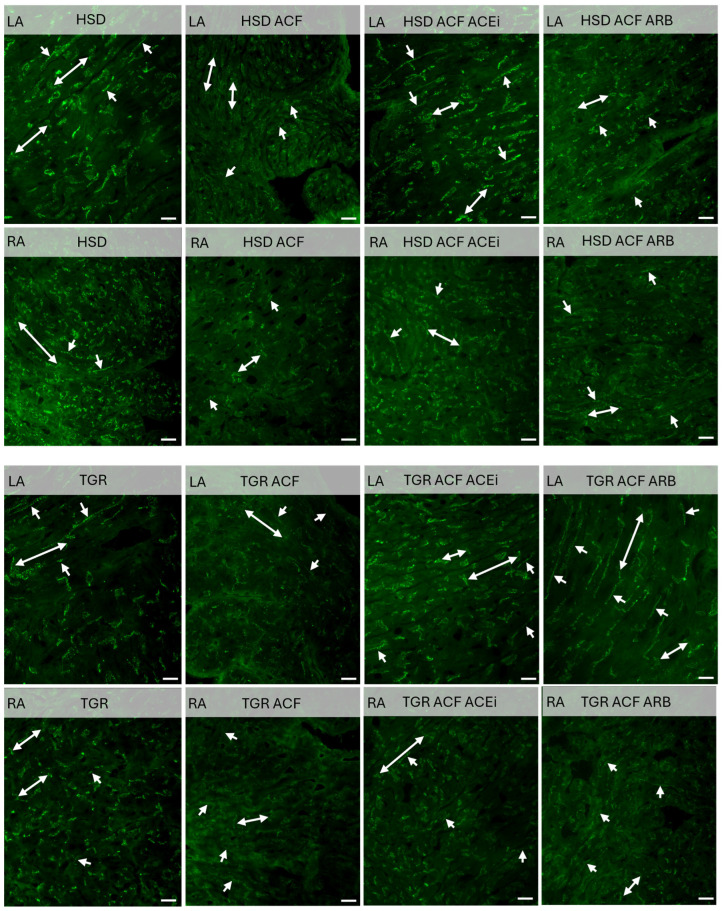
Immunolabeling of connexin-43 (green color) revealed higher expression predominantly at the intercalated disks (double arrows) in the left (LA) and right atria (RA) of normotensive HSD rats. In contrast, lower myocardial expression of Cx43 and its mislocalization to the lateral membranes (single arrows) were observed in hypertensive TGR. ACF-induced volume overload reduced Cx43 expression in both atria of HSD rats, while the reduction was less pronounced in TGR. Moreover, volume overload increased the localization of Cx43 to the lateral membranes of cardiomyocytes in HSD rats, whereas lateral localization persisted in hypertensive TGR. Treatment with ACEi or ARB increased Cx43 levels and attenuated its mislocalization in both atria of HSD rats, and to a lesser extent in TGR. HSD—Hannover Sprague Dawley rats; TGR—Ren-2 transgenic rats; ACF—aortocaval fistula, surgical model of volume overload; ACEi—treatment with the angiotensin-converting enzyme inhibitor, trandolapril; ARB—treatment with an angiotensin II type 1 (AT_1_) receptor blocker, losartan. The scale bar indicates 20 μm.

**Figure 6 biomolecules-15-01457-f006:**
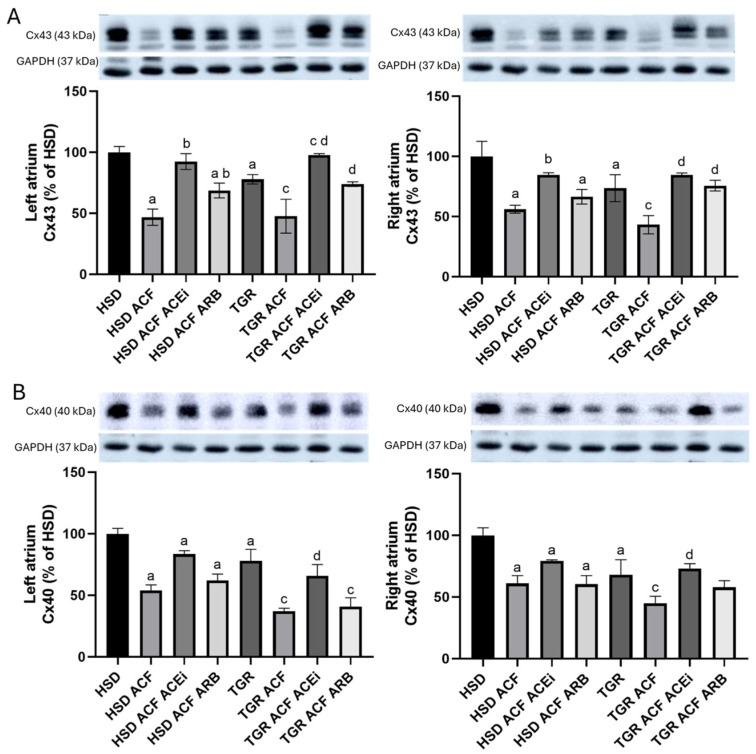
Protein levels of Cx43 (**A**) and Cx40 (**B**) normalized to GAPDH assessed by Western blot analysis. Hypertensive TGR rats exhibited lower levels of Cx43 and Cx40 in both the left and right atria compared to normotensive rats. A pronounced decrease in Cx43 and Cx40 protein levels was observed in both atria of normotensive HSD and hypertensive TGR rats due to volume overload. Treatment with an ACEi increased Cx43 and Cx40 levels (*p* > 0.05) in both atria of volume-overloaded normotensive and hypertensive rats. In contrast, ARB treatment increased Cx43 but did not affect Cx40 levels in either atrium of volume-overloaded rats. HSD—Hannover Sprague Dawley rats; TGR—Ren-2 transgenic rats; ACF—aortocaval fistula, surgical model of volume overload; ACEi—treatment with the angiotensin-converting enzyme inhibitor, trandolapril; ARB—treatment with an angiotensin II type 1 (AT_1_) receptor blocker, losartan. n = 10 per group. Data are presented as means ± SD; ^a^ *p* < 0.05 vs. HSD, ^b^ *p* < 0.05 vs. HSD ACF, ^c^ *p* < 0.05 vs. TGR, ^d^ *p* < 0.05 vs. TGR ACF. Western blot original images can be found in Appendix A.

**Figure 7 biomolecules-15-01457-f007:**
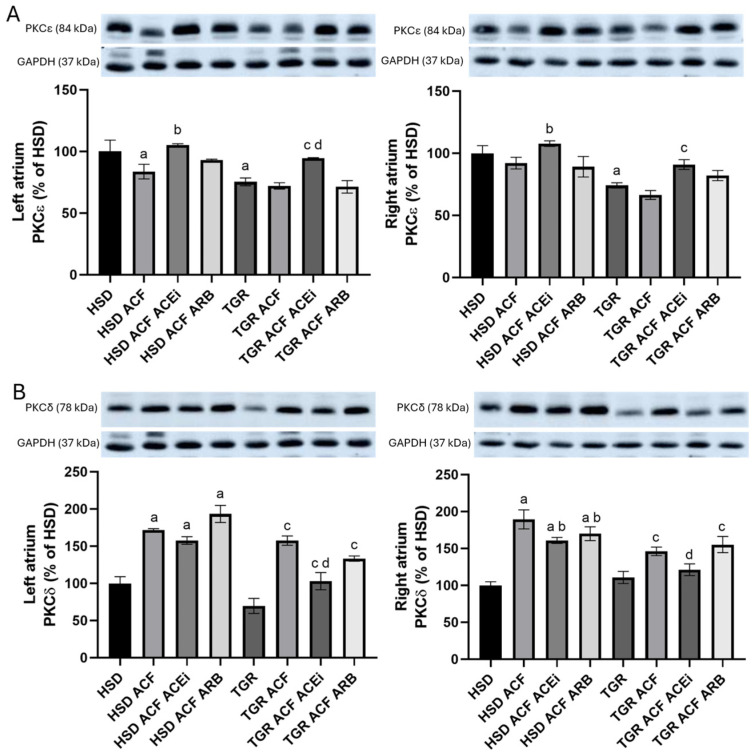
Protein levels of PKCε (**A**) and PKCδ (**B**) normalized to GAPDH assessed by Western blot analysis. Compared to normotensive HSD rats, PKCε levels were lower in both the left and right atria of hypertensive TGR rats. ACF-induced volume overload significantly reduced PKCε protein levels only in the left atrium of normotensive HSD rats. Treatment with an ACEi increased PKCε levels in both atria of volume-overloaded normotensive HSD and hypertensive TGR. PKCδ protein levels were significantly increased in both the left and right atria of normotensive HSD and hypertensive TGR rats in response to volume overload. Treatment with an ACE inhibitor reduced PKCδ levels only in the left and right atria of volume-overloaded hypertensive TGR rats. HSD—Hannover Sprague Dawley rats; TGR—Ren-2 transgenic rats; ACF—aortocaval fistula, surgical model of volume overload; ACEi—treatment with the angiotensin-converting enzyme inhibitor, trandolapril; ARB—treatment with an angiotensin II type 1 (AT_1_) receptor blocker, losartan. n = 10 per group. Data are presented as means ± SD; ^a^ *p* < 0.05 vs. HSD, ^b^ *p* < 0.05 vs. HSD ACF, ^c^ *p* < 0.05 vs. TGR, ^d^ *p* < 0.05 vs. TGR ACF. Western blot original images can be found in Appendix A.

**Figure 8 biomolecules-15-01457-f008:**
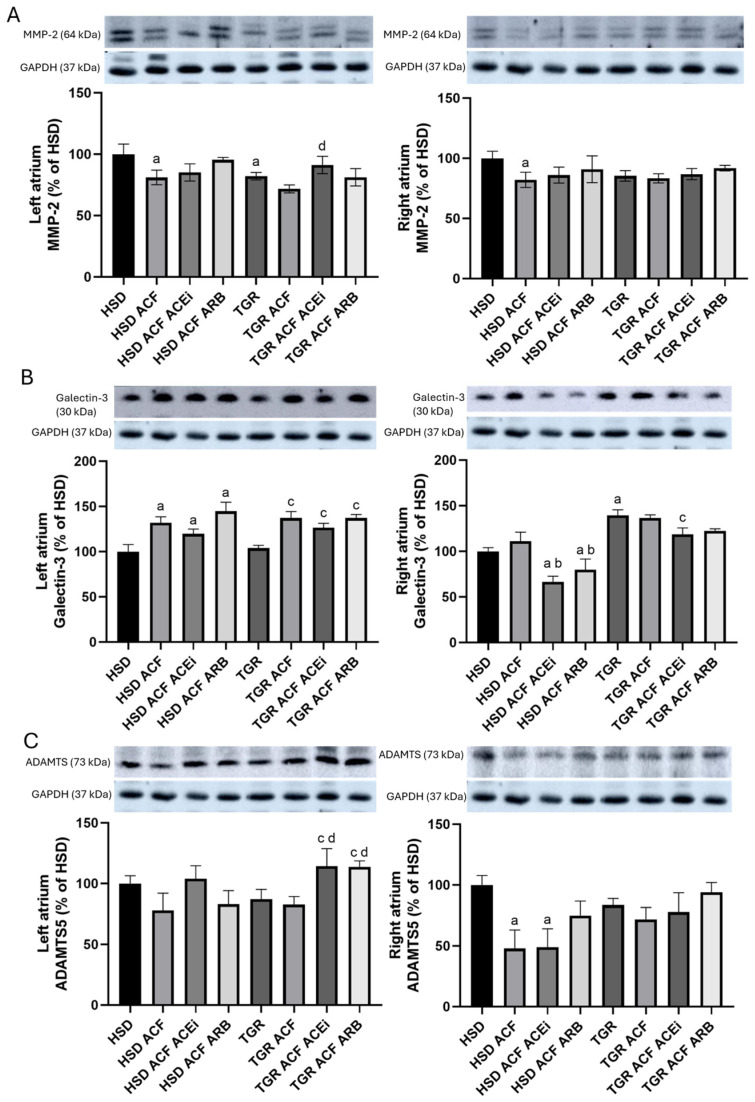
Protein levels of MMP-2 (**A**), Galectin-3 (**B**), and ADAMTS (**C**) normalized to GAPDH assessed by Western blot analysis. MMP-2 protein levels were significantly reduced in both the left and right atria due to volume overload, but only in normotensive HSD rats. Treatment with an ACEi increased MMP-2 levels in the left atrium of volume-overloaded TGR rats. Galectin-3 protein levels were increased in the left atrium in response to volume overload, regardless of treatment. Interestingly, Galectin-3 levels were significantly reduced in the right atrium of volume-overloaded HSD rats treated with either an ACEi or an ARB. In contrast to the left atrium, Galectin-3 levels were significantly increased in the right atrium of volume-overloaded TGR rats. ADAMTS protein levels were not affected by volume overload in the left atrium, but were decreased in the right atrium of normotensive HSD rats. Notably, treatment with either an ACE inhibitor or an ARB increased ADAMTS protein levels in the left atrium of volume-overloaded TGR rats. Data are presented as means ± SD; ^a^ *p* < 0.05 vs. HSD, ^b^ *p* < 0.05 vs. HSD ACF, ^c^ *p* < 0.05 vs. TGR, ^d^ *p* < 0.05 vs. TGR ACF. Western blot original images can be found in Appendix A.

**Table 1 biomolecules-15-01457-t001:** Biometric and biochemic parameters registered in experimental rats.

Parameters	HSD	HSD ACF	HSD ACF ACEi	HSD ACF ARB	TGR	TGR ACF	TGR ACF ACEi	TGR ACF ARB
BW (g)	543 ± 37	548 ± 45	537 ± 31	575 ± 56	652 ± 46 ^a^	483 ± 46 ^c^	606 ± 53 ^d^	643 ± 38 ^d^
HW (mg/mm)	40.5 ± 3.3	71.2 ± 5.2 ^a^	59.2 ± 4.7 ^ab^	63.8 ± 4.4 ^a^	52.0 ± 4.1 ^a^	76.2 ± 3.0 ^c^	51.5 ± 8.7 ^d^	65.2 ± 6.7 ^cd^
LVW (mg/mm)	18.8 ± 1.9	27.0 ± 2.8 ^a^	23.6 ± 3.2 ^a^	24. 7 ± 1.3 ^a^	22.2 ± 2.7	33.5 ± 3.0 ^c^	17.1 ± 3.5 ^d^	22.4 ± 1.8 ^d^
RVW (mg/mm)	5.2 ± 0.4	12.9 ± 1.9 ^a^	12. ± 1.2 ^a^	13.4 ± 1.34 ^a^	8.04 ± 1.3 ^a^	12.4 ± 0.4 ^c^	10.4 ± 1.6	13.5 ± 2.68 ^c^
Serum ANP (pg/mL)	2.7 ± 0.3	3.4 ± 0.3 ^a^	2.9 ± 0.5	2.3 ± 0.1 ^b^	2.4 ± 0.2	4.0 ± 0.7 ^c^	3.3 ± 0.2 ^d^	3.3 ± 0.9 ^d^
Serum TBARS (µmol/µg)	18.7 ± 0.8	25.0 ± 0.8 ^a^	19.9 ± 1.9 ^b^	20.7 ± 1.2 ^b^	21.3 ± 0.8	24.5 ± 1.1 ^c^	20.1 ± 0.9 ^d^	21.9 ± 2.1
Serum MMP-2 (%)	100 ± 8	95 ± 3	83 ± 12	96 ± 4	80 ± 7	65 ± 5	60 ± 12 ^c^	78 ± 4
Serum TIMP-2 (pg/mL)	245 ± 39	212 ± 2	341 ± 17 ^ab^	211 ± 20	171 ± 21	144 ± 20	226 ± 43 ^d^	134 ± 13

BW—body weight (g), HW—heart weight (g), LVW—left ventricular weight normalized to tibia length (mg/mm), RVW—right ventricular weight normalized to tibia length (mg/mm), ANP—Atrial natriuretic peptide (pg/mL), TBARS—thiobarbituric acid reactive substances (μmol/μg), MMP-2—Matrix metalloproteinase-2, TIMP-2—Tissue inhibitor of metalloproteinases-2 (pg/mL), HSD—Hannover Sprague Dawley rats; TGR—Ren-2 transgenic rats; ACF—aortocaval fistula, surgical model of volume overload; ACEi—treatment with the angiotensin-converting enzyme inhibitor, trandolapril; ARB—treatment with an angiotensin II type 1 (AT_1_) receptor blocker, losartan. n = 10 per group. Data are presented as means ± SD; ^a^ *p* < 0.05 vs. HSD, ^b^ *p* < 0.05 vs. HSD ACF, ^c^ *p* < 0.05 vs. TGR, ^d^ *p* < 0.05 vs. TGR ACF. These data were previously published in [7,11].

## Data Availability

Data are contained within the article and Appendix A.

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
