# Peer review of "Trandolapril Attenuates Pro-Arrhythmic Downregulation of Cx43 and Cx40 in Atria of Volume Overloaded Hypertensive and Normotensive Rats"

_biomolecules, 2025, doi:10.3390/biom15101457_

Round 1
Reviewer 1 Report
Comments and Suggestions for Authors
Interesting finding on the effect of trandolapril on Cx43 and Cx40 levels in right and left heart atria of normotensive as well as hypertensive volume overloaded rats. The data was obtained with ad hoc methodologies, and results are well presented and interpreted.
Minor issue
Some paragraphs of the Discussion section are very similar to the same topics described in the Results section. Could you please associate the heart cytoarchitecture changes with changes in levels of connexins?
Author Response
Reviewer 1
Interesting finding on the effect of trandolapril on Cx43 and Cx40 levels in right and left heart atria of normotensive as well as hypertensive volume overloaded rats. The data was obtained with ad hoc methodologies, and results are well presented and interpreted.
Minor issue
Some paragraphs of the Discussion section are very similar to the same topics described in the Results section. Could you please associate the heart cytoarchitecture changes with changes in levels of connexins? – diskusia
Dear Reviewer,
We would like to thank you very much for review our manuscript and we appreciate your relevant comments..
According to your suggestion we revised Discussion to avoid repetition of some paragraphs described in Results section as well as we discussed selected findings with coresponding available data in literature.
We investigated response of normotensive and hypertensive rat heart atria to volume overload induced by aortocaval fistula, focusing mainly on factors implicated in development of heart failure and arrhythmias. We detected that protein levels of connexin43 and conexin40, which are key factors involved in heart electrophysiology were significantly reduced in atria of pressure and volume overload rat hearts. These proteins located prevalently at the gap junctions channels in intercalated discs are esential for action potential propagation among cardiomyocytes. We and others have previously reported that reduced expression and/or abnormal topology of these proteins are arrhythmogenic due to disorders in action potential propagation, block of conduction thereby promoting myocardial electrical instability and mechanical dysfunction (Dhein 2021, Sykora 2023a, Egan Benova 2019, Tribulova 2015).
Structural remodeling including widened intermyocte spaces with abundant polymorpho-nuclear cells and patchy fibrosis in pressure or volume overloaded heart atria as observed in our experiment, are very likely implicated in the downregulation of connexins and their abnormal topology, thereby promoting electrophysiological disorders and electrical instability (Jarkovska 2021) promoting cardiac arrhythmias. Chronic pressure/volume overload of the atria facilitates AF, via enhanced extracellular matrix (ECM) accumulation manifested as tissue fibrosis (Li 2022) that deteriorates connexins topology and contribute to electrophysiological and contractile dysfunction in volume overloaded heart.(Jarkovska 2021).
The renin-angiotensin-aldosterone system (RAAS) governs myocardial collagen synthesis. Strong correlation between the abundance of patchy fibrosis and arrhythmia inducibility was found. Reduction of fibrosis-related arrhythmias by chronic renin-angiotensin-aldosterone system inhibitors were reported (Stein 2010). Chronic RAAS inhibition limited aging-related interstitial fibrosis. The lower arrhythmogeneity of treated mice was directly correlated to the reduced amount of patchy fibrosis. Fibrosis is a major contributor to heart failure and occurs in various pathological conditions including hypertension, congestive heart failure, post-infarction cardiac remodelling, etc. (Figueroa-Juarez 2025). Excessive deposition of extracellular matrix disrupts myocardial architecture and impairs the heart microvasculature leading to cardiac dysfunction.
RAS activation is associated with c-Src upregulation, Cx43 loss, reduced myocyte coupling, and arrhythmic sudden death. Those effects can be ameliorated by c-Src inhibition, which suggests that an increase in c-Src activity may participate in RAS-induced arrhythmias and that c-Src inhibitors might exert antiarrhythmic activity in states of RAS activation. Since we used a model of RAS activation limited to the heart, results may vary within vivo systemic RAS activation.
In a model of RAS activation, arrhythmic risk was correlated with reduced Cx43 amount and phosphorylation. RAS inhibition resulted in increased total and phosphorylated Cx43, decreased VT inducibility, and improved survival. (Iravanian,2011). Trandolapril exhibited a more pronounced effect which can be attributed to significant suppression of Angiotensin II plasma levels and c-Src kinase that is involved in Cx43 down-regulation and hampering cardiomyocyte communication, thereby providing a substrate for arrhythmia Sovari 2011. Traditionally, the effect of RAS on promoting atrial and ventricular arrhythmias is explained based on increased cardiac hypertrophy, fibrosis, and heterogeneity of the cardiac tissue.
Reviewer 2 Report
Comments and Suggestions for Authors
Reviewing the manuscript entitled, “Trandolapril attenuates pro-arrhythmic downregulation of Cx43 and Cx40 in atria of volume overloaded hypertensive and normotensive rats” by Sýkora M et al, this focuses on mechanisms of arrhythmias in volume overload-induced heart failure by the animal experiments. Although this is an interesting study, it is likely to involve many confounding factors, and it is difficult to fully evaluate the results.
The authors should describe cardiac function, blood pressure, and other factors that are considered to be the most important confounding factors in this study and thoroughly discuss their impact on the results.
Ren2 is a hypertensive rat model. If so, there should be a significant difference in blood pressure between the control group (HSD rat group). The authors should describe and interpret the results in terms of the possible reason for this difference.
Did both groups in the pathogenesis model develop heart failure?
Furthermore, did the pathogenesis model exhibit arrhythmia? If not, the tone of the authors' discussion is jumping to conclusions.
RAS inhibitors work to suppress cardiac fibroblast proliferation in heart failure and improve prognosis. However, as they are antihypertensive drugs, the changes in blood pressure in the four TGR groups are extremely important, but this is not described.
In the TGR ACF group, it appears that not only volume overload but also pressure overload is an extremely important confounding factor. Is this correct?
The TGR group was clearly heavier than the HSD group, and only the TGR ACF group showed significant weight loss. Is it possible that there were other confounding factors besides hemodynamics that were causing the abnormalities?
The authors should discuss in detail the differences in the results between ACEi and ARB. Why did these differences arise when both RAS inhibitors were used?
Author Response
Reviewer 2
Reviewing the manuscript entitled, “Trandolapril attenuates pro-arrhythmic downregulation of Cx43 and Cx40 in atria of volume overloaded hypertensive and normotensive rats” by Sýkora M et al, this focuses on mechanisms of arrhythmias in volume overload-induced heart failure by the animal experiments. Although this is an interesting study, it is likely to involve many confounding factors, and it is difficult to fully evaluate the results. Yes, we totally agree with you.
Dear reviewer,
Thank you very much for reviewing our manuscript and relevant comments, ideas and suggestions. We responded majority of them and included in revised manuscript. Regarding unanswered comments, questions or ideas, we will pay more attention in our further experiment.
The authors should describe cardiac function, blood pressure, and other factors that are considered to be the most important confounding factors in this study and thoroughly discuss their impact on the results.
Ren2 is a hypertensive rat model. If so, there should be a significant difference in blood pressure between the control group (HSD rat group). The authors should describe and interpret the results in terms of the possible reason for this difference.
In agreement with your comment we included information about blood pressure and cardiac function assessed in this experiment and recently published (Kratky 2021). Therefore, we can pay more attention to this issue in our further experiment.
Did both groups in the pathogenesis model develop heart failure?
Yes, both normotensive and hypertensive rats develop heart failure due to volume overload as it was assessed by echocardiography from the same experiment but investigating renal function and published by Krátky et al. 2021 (10.1038/s41598-021
Furthermore, did the pathogenesis model exhibit arrhythmia? If not, the tone of the authors' discussion is jumping to conclusions.
Cardiac arrhythmias were not examined/observed during experiments. There is an intention to examine AF inducibility in further experiments with sufficient number of rats per group. However, available data refer proarrhythmic electrophysiological remodeling due to aortocaval induced volume overload and heart failure (e.g. Jarkovska 2021). In a model of RAS activation, arrhythmic risk was correlated with reduced Cx43 amount and phosphorylation. RAS inhibition resulted in increased total and phosphorylated Cx43, decreased VT inducibility, and improved survival, Iravanian 2011, doi:10.1007/s00109-011-0761-3.
We aimed to investigated response of normotensive and hypertensive rat heart atria to volume overload induced by aortocaval fistula, focusing mainly on factors implicated in development of heart failure and atrial arrhythmias. We detected that protein levels of connexin43 and conexin40, which are key factors involved in heart electrophysiology were significantly reduced in atria of pressure and volume overload rat hearts. These proteins located prevalently at the gap junction’s channels in intercalated discs are essential for action potential propagation among cardiomyocytes. We and others have previously reported that reduced expression and/or abnormal topology of these proteins are arrhythmogenic due to disorders in action potential propagation, block of conduction, thereby promoting myocardial electrical instability and mechanical dysfunction (Dhein 2021, Sykora 2023a, Egan Benova 2019, Tribulova 2015).
RAS inhibitors work to suppress cardiac fibroblast proliferation in heart failure and improve prognosis. However, as they are antihypertensive drugs, the changes in blood pressure in the four TGR groups are extremely important, but this is not described.
Thank you for this comment. According to our previous findings, Krátky et al. 2021, comparing to nontreated rats, mean arterial pressure was significantly reduced due to volume overload in hypertensive while not in normotesive rats. Treatment with RAS inhibitors did not affect significantly blood pressure in normotensive volume overloaded rats unlike hypertensive rats in which treatment slightly reduced blood pressure comparing to nontreated volume overloaded rats.
We would like to note that chronic pressure/volume overload causes cell stretch that is generally considered to promote fibrosis by directly activating fibroblasts of the atria that facilitates AF, via enhanced extracellular matrix (ECM) accumulation manifested as tissue fibrosis (Li 2022, doi:10.1093/cvr/cvab035) which deteriorates connexins expression and topology thereby contribute to electrophysiological and contractile dysfunction in pressure and volume overloaded heart (Jarkovska 2021 10.3389/fphar.2021.729568). Clarity in this area is needed to improve our understanding of AF pathophysiology and assist in therapeutic development.
In the TGR ACF group, it appears that not only volume overload but also pressure overload is an extremely important confounding factor. Is this correct?
Yes, we think so. However, this issue requires further investigations and supporting evidence.
The TGR group was clearly heavier than the HSD group, and only the TGR ACF group showed significant weight loss. Is it possible that there were other confounding factors besides hemodynamics that were causing the abnormalities?
The TGR strain displays a chronically overactivated renin–angiotensin system and, under baseline conditions, exhibits higher body mass compared with normotensive controls, likely due to Ang II–mediated anabolic effects and increased appetite, which promote greater accumulation of fat and lean mass. The induction of an aortocaval fistula (ACF) produces chronic volume overload and heart failure, which in turn triggers systemic inflammation, catabolic pathways, and increased energy expenditure (Abassi, 2011 - 10.1155/2011/729497). In TGR, this hemodynamic stress acts on a background of Ang II–driven pro-inflammatory, proteolytic, and metabolic alterations, thereby amplifying the maladaptive response. Consequently, TGR subjected to ACF experience due to decompensated heart failure, a rapid and pronounced loss of body mass, consistent with cardiac cachexia, in contrast to the higher baseline body mass observed in TGR without ACF.
The authors should discuss in detail the differences in the results between ACEi and ARB. Why did these differences arise when both RAS inhibitors were used?
The difference between these drugs lies primarily in their mechanisms of action. Trandolapril, an ACE inhibitor (ACEi), inhibits the activity of the angiotensin-converting enzyme, a key component of the renin-angiotensin system that converts angiotensin I into angiotensin II. Thus ACE inhibitors lead to a reduction in circulating levels of angiotensin II, which can itself directly activate or inactivate pathways of interest. For example, angiotensin II is involved in the down-regulation of Connexin-43 (Cx43) (Chen, 2020 - 10.33549/physiolres.934488, Cai, 2006 - 10.1631/jzus.2006.B0648). In contrast, Losartan, an angiotensin receptor blocker (ARB), does not reduce the level of circulating angiotensin II, it acts solely by blocking the AT₁ receptors. Any non-receptor mediated functions of angiotensin II remain unchanged.
Reviewer 3 Report
Comments and Suggestions for Authors
Sýkora et al present immunohistochemical and biochemical data examining the effect of Trandolapril and losartan in atria of volume overloaded hypertensive and normotensive rats with a focus on Connexin 40 and 43. While the study addresses an interesting topic, substantial restructuring and additional clarification are needed.
In line 364 of the discussion, the authors state that the recorded biometric parameters have already been published in a previous study (Ref. 7). This information should be mentioned at the beginning of the Results section, where biometric parameters are first introduced.
Lines 375–403 in the discussion section describe the results shown in Figures 1–4. As this is a direct presentation of results rather than interpretation, it should be moved to the Results section.
The immunohistochemical staining presented in Figures 1–4 should be accompanied by quantification of the staining to substantiate the conclusions drawn. Without quantitative data, the statements remain descriptive rather than conclusive.
For Figures 6–8, different symbols should be used to indicate statistically significant differences versus the HSD group and versus the TG group. Using * for both is confusing and may lead to misinterpretation.
The authors note that losartan treatment produced distinct effects on biometric parameters in volume-overloaded normotensive rats compared to hypertensive rats. The rationale for this difference is not discussed. A detailed explanation or interpretation of this finding should be included in the discussion.
Based on which findings do the authors conclude that the right atris of normotensive HSD rats and hypertensive TGR rats are more susceptible to volume overload?
Overall, the discussion section largely restates the results rather than analyzing them in the context of existing literature. The section should be rewritten to provide deeper interpretation, linking the findings to known mechanisms and prior studies in the field.
Comments on the Quality of English LanguageLine 334, 'an' can be removed.
Line 39 -42, sentence does not make sense.
Line 352, compared instead of comparing
Author Response
Reviewer 3
Sýkora et al present immunohistochemical and biochemical data examining the effect of Trandolapril and losartan in atria of volume overloaded hypertensive and normotensive rats with a focus on Connexin 40 and 43. While the study addresses an interesting topic, substantial restructuring and additional clarification are needed.
Dear Reviewer,
We would like to thank you very much for review our manuscript and we appreciate your relevant comments and suggestions.
We addressed your comments in revised manuscript.
In line 364 of the discussion, the authors state that the recorded biometric parameters have already been published in a previous study (Ref. 7). This information should be mentioned at the beginning of the Results section, where biometric parameters are first introduced.
This information was included at the beginning of the Results section, at the end of subsection 3.1 (Biometric and biochemical parameters).
Lines 375–403 in the discussion section describe the results shown in Figures 1–4. As this is a direct presentation of results rather than interpretation, it should be moved to the Results section.
Yes, we did it.
The immunohistochemical staining presented in Figures 1–4 should be accompanied by quantification of the staining to substantiate the conclusions drawn. Without quantitative data, the statements remain descriptive rather than conclusive.
Thank you for your comments regarding the enrichment of our results through quantification of the histological and histochemical images. Unfortunately, we were not able to perform quantification due to an insufficient number of samples per group for statistical analysis. As mentioned in the Methods section, we included 10 animals per group and used all 10 hearts for Western blot analysis to ensure statistical significance. For the microscopic analyses, however, we selected only 3 hearts per group as a representative subset to confirm the patterns observed in the Western blot experiments. This approach allowed us to validate the trends without processing all samples. We acknowledge that quantification would provide stronger support for our claims; however, the qualitative observations from the microscopic images are consistent, informative, and align with the quantitative Western blot results, thus providing meaningful evidence for the trends we report. Moreover, our aim was also to evaluate changes in the distribution of fibrosis (interstitial / compact / heterogeneous) and enzymatic activity (in the endocardial vs. epicardial regions), which cannot be fully captured by quantitative intensity analysis.
For Figures 6–8, different symbols should be used to indicate statistically significant differences versus the HSD group and versus the TG group. Using * for both is confusing and may lead to misinterpretation.
We have revised the symbols for significant differences in the Table and in all Figures.
The authors note that losartan treatment produced distinct effects on biometric parameters in volume-overloaded normotensive rats compared to hypertensive rats. The rationale for this difference is not discussed. A detailed explanation or interpretation of this finding should be included in the discussion.
We are not sure whether we understand your comment because we did not compare biometric parameters after treatment with Losartan in volume-overloaded normotensive rats (HSD ACF ARB) with hypertensive rats (TGR ACF ARB).
Based on which findings do the authors conclude that the right atria of normotensive HSD rats and hypertensive TGR rats are more susceptible to volume overload?
As we can see in Figure 1, polymophonuclears are more abundant in right versus left atria of normotensive as well as hypertensive volume overloaded rats. Besides, as demonstrated in Figure 2 extracellular collagen deposition was much increased in right versus left atria of normotensive as well as hypertensive volume overloaded rats. In Figure 3, alkaline phosphatase activity differ in the right versus left atria and in Figure 4 the difference was evident in dipeptidyl peptidase activity.
Overall, the discussion section largely restates the results rather than analysing them in the context of existing literature. The section should be rewritten to provide deeper interpretation, linking the findings to known mechanisms and prior studies in the field.
Yes, we did revision according to your suggestion.
Round 2
Reviewer 2 Report
Comments and Suggestions for Authors
This is an acceptable quality. Congrats.
Author Response
Thank you very much for your encouraging response.
Reviewer 3 Report
Comments and Suggestions for Authors
The authors have addressed most of my questions. I would, however, like to reiterate that without quantification and relying solely on representative images, the conclusions should be considered preliminary. If quantification cannot be provided, this should be acknowledged as a limitation of the study, noting that the findings need to be interpreted with caution.
Author Response
Thank you for your valuable comment. We have addressed your suggestion and added the requested statement regarding the lack of quantification as a limitation in the revised version of the manuscript, under the section Limitations of the study.
'Another limitation of this study is the absence of quantitative analysis for microscopic findings. As the conclusions are based on representative images, they should be interpreted with caution. Further studies including quantitative assessment are required to validate these observations.'